

# International trade market forecasting and decision-making system: multimodal data fusion under meta-learning

Yiming Bai[1] and Muhammad Asif[2]

[1] Faculty of Social Science & Public Policy, Department of International Development, King's College London, London, United Kingdom
[2] Department of Computer Science, National Textile University, Faisalabad, Pakistan

Corresponding author
Yiming Bai, ttaa202411@163.com

## ABSTRACT

Traditional market analysis tools primarily rely on unidimensional data, such as historical trading records and price trends. However, these data are often insufficient to reflect the actual state of the market fully. This study introduces a meta-learning-based (MLB) multimodal data fusion approach to optimize feature extraction and fusion strategies, addressing the complexity and heterogeneity inherent in international trade market data. Initially, the mel-frequency cepstral coefficients (MFCC) method is employed to transform the original audio signal into more discriminative spectral features. For image data, the convolutional block attention module (CBAM) is incorporated to capture both channel-wise and spatial attention, thereby improving the model's ability to focus on market-relevant information. In the feature fusion stage, a meta-learning bidirectional feature pyramid network (ML-BiFPN) is proposed to refine the interaction of multi-scale information *via* a bidirectional feature pyramid structure. An adaptive weighting mechanism is employed to adjust the feature fusion ratio dynamically. Experimental results demonstrate that the proposed multimodal data fusion model, ML-BiFPN under meta-learning, significantly outperforms existing methods in prediction performance. When tested on the publicly available Trade Map dataset, the average accuracy improves by 9.37%, and the F1-score increases by 0.0473 compare to multilayer perceptron (MLP), achieving a prediction accuracy of 94.55% and an F1-score of 0.912. Notably, under small sample conditions, the model's advantage becomes even more pronounced, with an average precision (AP) improvement of 2.79%. These findings have significant implications for international trade market forecasting and decision-making, providing enterprises with a more comprehensive understanding of market dynamics, enhancing forecasting accuracy, and supporting scientifically informed decision-making to gain a competitive edge in the marketplace

## INTRODUCTION

In today's increasingly globalized world, the international trade market presents a complex and dynamic landscape. To remain competitive in this challenging environment, enterprises must accurately predict market trends and make scientifically grounded

decisions. In the actual workflow of international trade, audio data, such as voice call recordings between trading parties and audio files of meeting discussions, constitute a vital part of daily communication. These audio materials can authentically reflect the tone, emotions, and key points conveyed during interactions, making them of utmost importance for analyzing negotiation dynamics and identifying cooperation intentions. Image data, including product photographs, on-site inspection images, and monitoring videos of the transportation process, are also indispensable. They offer a visual representation of product quality, production environments, and logistics status, which holds significant value for assessing transaction risks and ensuring transaction security.

Single-dimensional data struggles to capture the whole dynamics of the market. Particularly when confronted with emergencies, policy changes, or shifts in consumer preferences, the predictive accuracy of traditional methods is significantly compromised. Moreover, the heterogeneity and noise issues among different data sources pose challenges to data fusion. With the rapid advancements in big data and artificial intelligence, multimodal data fusion has emerged as a crucial solution to address this gap. Multimodal data fusion (*Xia et al., 2023*) involves the integration and analysis of data from diverse channels, each with distinct formats and characteristics, to uncover hidden patterns and insights. In the context of international trade, these data sources may include news reports, social media posts, product images, video advertisements, audio recordings and voice analysis. Each modality provides valuable market insights, including consumer preferences, competitor behaviors, and shifts in the policy landscape. By effectively combining these multimodal data, enterprises can gain a more holistic understanding of market dynamics, thereby improving the accuracy of market forecasting and enhancing decision-making processes (*Li et al., 2024*).

Meta-learning (*Vettoruzzo et al., 2024*), a subset of machine learning, focuses on improving a model's ability to generalize by learning the process of learning itself. The integration of meta-learning into international trade market forecasting and decision-making systems can significantly enhance the adaptability and robustness of these systems (*Vettoruzzo et al., 2024*; *Gharoun et al., 2024*). By leveraging historical data and prior experiences, meta-learning enables models to rapidly adjust to new market conditions and diverse data types, thereby facilitating precise predictions and flexible decision-making.

Despite the promising potential of multimodal data fusion and meta-learning in international trade market forecasting, several challenges remain (*Bartolucci et al., 2023*). On the one hand, the collection, processing, and storage of multimodal data entail high costs and require advanced technical infrastructure. On the other hand, the heterogeneity and noise inherent in different data modalities pose significant challenges to the effective fusion of data. Additionally, while research on meta-learning has primarily focused on enhancing model adaptability and generalization, there is a lack of systematic exploration into how to design more effective meta-learning algorithms and frameworks tailored to the specific characteristics and needs of international trade markets. For instance, developing more robust training strategies and learning objectives based on historical trade data

remains a key challenge for achieving more accurate market predictions and providing actionable decision support.

To design an effective model training strategy and learning objectives for more efficient and accurate meta-learning algorithms in international trade market forecasting and decision-making systems, This study employs a meta-learning-based multimodal data fusion approach. It proposes efficient methods for audio and image feature extraction, enabling multimodal data fusion within the meta-learning framework, and constructs a SwinTransformer-based YOLO framework. These innovations offer new ideas and approaches for market forecasting and decision-making in international trade. The specific contributions of this study are as follows:

(1) Efficient audio and image feature extraction: This article leverages mel-frequency cepstral coefficients (MFCC) for audio feature extraction and the convolutional block attention module (CBAM) for image data. By incorporating channel and spatial attention mechanisms, the model enhances its focus on crucial channels and regions, thereby improving the accuracy of market predictions.

(2) Multimodal data fusion within the meta-learning framework: A meta-learning framework is introduced to facilitate the fusion of multimodal data, including diverse data sources such as news reports, audio, and video. By integrating these varied data types, the model enhances both the accuracy of market forecasting and the scientific rigor of decision-making processes.

(3) SwinTransformer-based You Only Live Once (YOLO) framework: The Swin Transformer is employed as the backbone network to extract features from multimodal data within the meta-learning framework efficiently. Additionally, the feature fusion strategy is optimized to enhance the model's prediction accuracy, particularly in conditions with small sample sizes.

This article will introduce the current research status of multimodal data analysis and prediction models, as well as the research gaps in data prediction models under the meta-learning framework, in "Related Works". "Materials and Methods" presents the multimodal data feature processing methods in this article, as well as the a meta-learning bidirectional feature pyramid network (ML-BiFPN) model for fusing audio and image data within the meta-learning framework. "Experiments and Analysis" will conduct experimental analyses, examining the performance of feature extraction and the predictive performance of different multimodal data fusion models under the meta-learning framework with varying sample sizes, in order to analyze their impact on the decision-making system. Finally, in "Conclusion and Limitations", the performance of the ML-BiFPN model will be analyzed, along with potential future research directions.

## RELATED WORKS

Audio and image data fusion involves extracting features from different data sources and transforming them into a uniform format for fusion operations (*Yang, Li & Huang, 2024*; *Islam et al., 2023*). This technique integrates the interrelated information from audio and image data to compensate for their limitations, thereby enhancing the comprehensiveness and accuracy of the information and improving the precision and reliability of data

analysis. Traditional audio and image data fusion methods include feature-level fusion (*Zhang, Yang & He, 2022*; *Islam et al., 2023*), decision-level fusion (*Jin et al., 2022*; *Singh et al., 2022*; *Li et al., 2023*), and hybrid fusion (*Liu et al., 2025*). With the rapid development of deep learning and computing, audio and image data fusion methods based on deep learning and attention mechanisms have become widely adopted in areas such as sentiment analysis and data prediction (*Hosseini, Yamaghani & Arabani, 2024*).

In recent years, with the increasing prevalence of multimedia content on social media, multimodal data predictive analytics has emerged as a prominent research direction within the field of data prediction. Compared to unimodal data predictive analytics, multimodal data predictive analytics enhances the accuracy and robustness of data predictions by integrating multiple data sources, allowing for a more comprehensive consideration of various information types, such as text, image, and audio. *Hosseini, Yamaghani & Arabani (2024)* advances research in multimodal sentiment prediction by employing multi-kernel learning based on text, audio, and visual modalities for sentiment analysis. In earlier fusion approaches, the quality of the fusion results was influenced not only by the fusion method but also by the quality of the extracted features. *Ning et al. (2025)* proposes decomposing video frames into multiple unimodal data types, such as audio and image data, assigning unclassified weights to data with varying levels of distinctness using the attention mechanism and subsequently fusing the classification results from different modules to obtain the final predictive outcomes. *Wu (2024)* combines extracted audio and image features and then inputs the fused features into a support vector machine for final classification. *Qureshi et al. (2022)* develops unimodal sentiment classifiers for audio and video separately, fine-tunes them using sentiment datasets, and then removes the unimodal SoftMax classifier. A multimodal classifier is applied after cascading the features from both modalities for final sentiment classification, demonstrating that using the unimodal sentiment classifier before feature cascading allows for the extraction of more detailed features, which ultimately facilitates effective multimodal data fusion.

The Kalman filter is also employed as one of the fusion classifier methods in late fusion techniques. *Khurana et al. (2023)* utilizes a basic classifier to obtain predicted scores for video sequences, which are scaled between 0 and 1 and then fuses these predicted scores using a Kalman filter. *Khurana et al. (2023)* utilizes convolutional neural networks to automatically learn from raw audio and image data, representing them and training multiple support vector machine (SVM) classifiers to make local decisions based on various features. Different weights are assigned to the decision results from numerous classifiers, and the final decision is obtained through weighted summation. *Cai et al. (2024)* proposes a multilevel hybrid fusion architecture for dynamic Bayesian networks, exploring audiovisual recognition through the fusion of audio and image data. This approach combines model-level and decision-level fusion to enhance the accuracy of audiovisual recognition. *Wu et al. (2023)* introduces a novel attention-based multimodal fusion strategy that assigns varying weights to different modalities based on current input features and recent historical information, thereby improving model performance.

In addition, in recent years, artificial intelligence (AI) has also made remarkable progress in the fields of economic forecasting and international trade analysis. For instance, *Liu & Long (2020)* proposed a deep-learning-based time-series forecasting model for financial market trend analysis, while *Seddigh, Shokouhyar & Loghmani (2023)* explored the application of business intelligence systems in supply chain management. These studies have provided valuable references for our research, particularly in terms of applying artificial intelligence (AI) technologies to market forecasting and decision-making in international trade.

However, international trade market data is highly complex, dynamic, and uncertain, with the correlations and interactions between data from different modalities often difficult to capture accurately. Furthermore, the distribution and characteristics of these data may undergo significant changes over time, making it challenging for traditional fusion models to adapt to such fluctuations, which may result in suboptimal fusion outcomes. Additionally, the multimodal fusion of data in international trade markets is particularly hindered by the issue of small sample sizes. In the context of deep learning, meta-learning techniques have demonstrated relatively effective detection performance even with small sample sizes. When faced with limited levels and parameters for sample optimization structures, meta-learning can effectively learn text, image, or audio features from small samples, achieving significant learning performance. Moreover, under the condition of comparable detection accuracy, a single-stage model generally exhibits lower complexity and faster detection speed compared to a two-stage meta-learning-based detection framework.

## MATERIALS AND METHODS

The relevance of voice call data in trade decision-making lies in its ability to capture subtle emotional changes, cooperation intentions, and potential risks during communication. For instance, through sentiment analysis, we can determine whether the other party is enthusiastic about the cooperation or if they have doubts and concerns. This enables us to adjust our negotiation strategies accordingly. Product images, on the other hand, are directly related to the quality and appearance of the goods, serving as an essential basis for evaluating the transaction value. By utilizing image processing technologies, we can detect product defects, assess their market competitiveness, and make procurement or sales decisions based on this information.

In terms of usability, the advancement of technology has made the collection, storage, and processing of audio and image data more convenient. Modern communication tools make it easy to obtain voice call recordings, while high-definition cameras and image processing software render the collection and analysis of product images feasible. Therefore, incorporating audio and image modalities into the trade decision-making process is both practically feasible and of great value.

In this section, to address the issue of poor prediction performance when sample sizes are limited in international trade market prediction systems, SwinTransformer is adopted as the backbone network model to enhance the extraction of global feature information.

Additionally, a meta-learning-based YOLO framework is proposed, and multimodal data fusion is integrated within the model to improve prediction accuracy and performance.

## Data preprocessing

During the construction of the trade map dataset, we meticulously documented the generation and labeling methods for multimodal elements (including audio and images) to ensure the reproducibility of research results.

Audio data generation and labeling: Audio data primarily originates from trade-related voice call recordings, conference calls, and other similar sources. To protect privacy, all audio has undergone desensitization processing. Professional teams then transcribe and label these audio files. The labeling content includes conversation topics, emotional tendencies, key decision-making points, and other relevant information for subsequent analysis.

Image data generation and labeling: Image data encompasses product photos, screenshots from transportation process monitoring videos, and other relevant images. After preprocessing (such as cropping, scaling, and normalization), these images are labeled by domain experts. The labeling content includes product categories, quality grades, transportation statuses, and other relevant information.

To guarantee the consistency and accuracy of labeling, we adopted a multi-person labeling and cross-validation approach. In this experiment, we divided the Trade Map dataset into a training set, a validation set, and a test set. Specifically, we used 70% of the data as the training set for model training, 15% of the data as the validation set for parameter adjustment and performance monitoring during the model training process, and the remaining 15% of the data as the test set for evaluating the model's final performance.

## Multimodal feature processing

In the international trade market system, both audio and image data play a crucial role. Audio data primarily consists of voice call records between transaction parties, meeting discussion audio, and similar content, which can effectively reflect the tone, emotion, and focus of the communication process. This data is valuable for analyzing the negotiation dynamics and capturing the intent to cooperate. In contrast, image data includes product photos, site inspection images, and transportation process monitoring videos, which provide visual insights into product quality, production environments, and logistics conditions. These elements are crucial for evaluating transaction risks and ensuring transaction security.

### *MFCC-based audio feature processing*

Raw audio data is a digital signal captured in the form of sound, typically represented as a waveform, and often contains significant amounts of redundant and noisy information. In

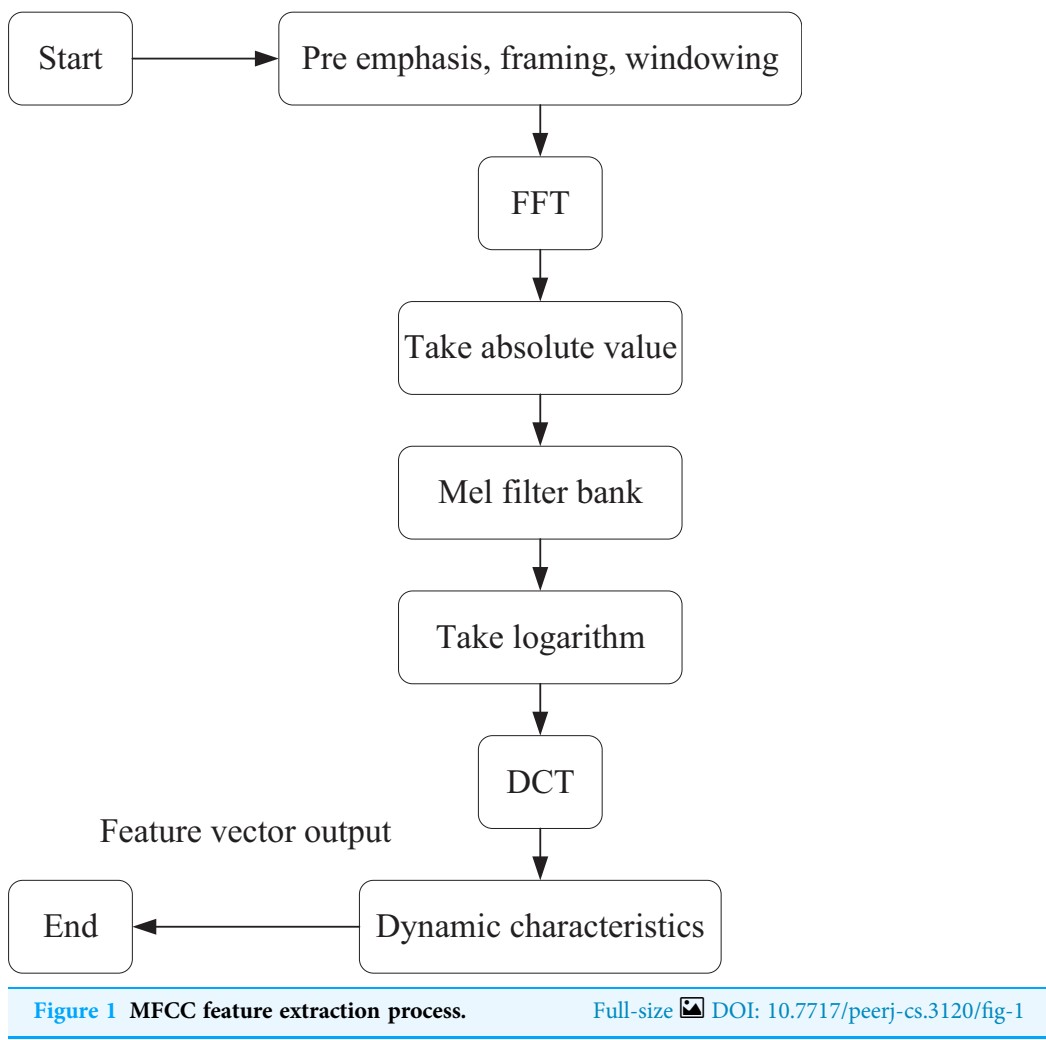

**Figure 1 MFCC feature extraction process.**

this article, MFCC feature extraction is applied to the audio data to facilitate the distinction of high-dimensional features. The specific process is illustrated in Fig. 1.

The first step is pre-emphasis, which involves applying a first-order high-pass filter to the signal in the time domain. This process increases the high-frequency energy by amplifying the high-frequency components of the signal while attenuating the low-frequency components. The goal is to enhance the high-frequency signals and improve the signal-to-noise ratio, as follows:

$$H(z) = 1 - \alpha z^{-1} \tag{1}$$

where $\alpha$ is the pre-emphasis factor and $0.9 < \alpha < 1$.

Segmentation then divides the signal into short time windows, processing each frame independently. This step ensures that the signal remains relatively stable over short periods

in the time domain. The Hanning window is applied so that after sub-framing and windowing, the original signal $x(n)$ is transformed into:

$$g(n) = x(n)w(n). \tag{2}$$

After applying the window, each frame of the time-domain signal is converted into a frequency-domain signal through the Fourier transform. This transformation facilitates the extraction of frequency-domain features. At this stage, the time-domain signal $g(n)$ is transformed into the frequency-domain signal $G(k)$:

$$G(k) = \sum_{n=0}^{N-1} g(n)e^{-j\frac{2\pi}{N}nk} \tag{3}$$

where $N$ denotes the number of points in the FFT, transform and $k$ represents the frequency index. Next, the frequency-domain signal $G(k)$ is expressed in complex form as follows:

$$G(k) = \alpha \cos\theta_k k + j\sin\theta_k k = \alpha_k + jb_k \tag{4}$$

$$E(k) = \alpha_k^2 + b_k^2. \tag{5}$$

A Mel filter bank is then applied to map the signal onto the Mel frequency axis, simulating the auditory characteristics of the human ear. The purpose of the Mel filter bank is to weight and sum the signals along the frequency axis according to specific weights, effectively dividing the frequency axis into several frequency bands. The Mel filter energy center frequency is defined as:

$$f(m) = \frac{N}{F_s}B^{-1}\left(B(f_l(m)) + m\frac{B(f_h(m)) - B(f_l(m))}{M+1}\right) \tag{6}$$

where $f(m)$ denotes the center frequency of the $m$-th filter, and $B(f)$ represents the Mel frequency. The upper and lower frequencies of the triangular filter $m$ are denoted accordingly.

Finally, the discrete cosine transform (DCT) is applied to compress and downsize the energy coefficients of the Mel frequency, thereby reducing the dimensionality of the MFCC feature vector. By discarding high-frequency coefficients, we can also achieve a denoising effect. The resulting coefficients are obtained as:

$$c(m) = \sqrt{\frac{2}{M}\sum_{n=1}^{M} S(m)\cos\left(\frac{\pi m(n-0.5)}{M}\right)}, 1 \leq m \leq L \tag{7}$$

where $L$ is the dimension of the MFCC feature, $c(m)$ denotes the $m$-th dimension of the MFCC feature, and:

$$s(m) = \ln\left(\sum_{k=0}^{N-1} E(k)H_m(k)\right) \tag{8}$$

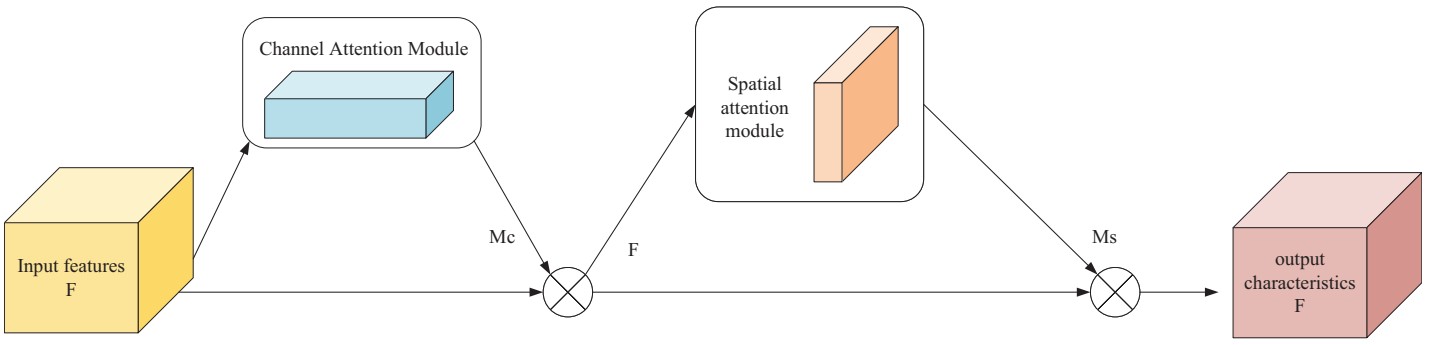

**Figure 2 CBAM structure.**

$$H_m(k) = \begin{cases} 0, k < f(m-1) \ or \ k > f(m+1) \\ \frac{2(k-f(m-1))}{(f(m+1)-f(m-1)(f(m)-f(m-1)))}, f(m-1) \le k \le f(m) \\ \frac{2(f(m+1)-k)}{(f(m+1)-f(m-1)(f(m+1)-f(m-1)))}, f(m) \le k \le f(m+1) \end{cases} . \tag{9}$$

### CBAM-based image feature processing

In this section, the channel and spatial attention mechanism is introduced in the form of the CBAM. Specifically, the feature map $F$ with input size $C \times H \times W$ is multiplied by the channel attention map $M_c$ of $C \times 1 \times 1$ to get the feature map F′, and then multiplied by the spatial attention map $M_s$ of $1 \times H \times W$ to get the final output feature map F″.

As shown in Fig. 2, CBAM is primarily composed of two components: the channel attention module (CAM) and the spatial attention module (SAM).

The input feature map F of channel attention map $M_c$ of size $C \times H \times W$ is subjected to global maximum pooling and average pooling to aggregate the spatial information of the feature maps, respectively, and two $C \times 1 \times 1$ feature maps $F_{max,c}$ and $F_{avg,c}$ are obtained, which are transferred to a multilayer perceptron (MLP) operation with only one hidden layer to perform a summation operation on the output features. Finally, the weight coefficients of each channel are obtained after Sigmoid activation, *i.e.*, the channel attention feature map $M_c$. The channel attention map is computed as follows.

$$M_c(F) = \sigma(W_1(W_0(F_{avg,c})) + W_1(W_0(F_{max,c}))) \tag{10}$$

where $\sigma$ is the Sigmoid function and $W_0 \in \mathbb{R}^{C/r \times C}, W_1 \in \mathbb{R}^{C/C \times r}$ denotes the weight of MLP respectively. CBAM applies attention mechanisms in both the channel and spatial domains. Therefore, the input to the SAM is the feature map $F'$, which is obtained by multiplying the channel attention feature map with the original feature map. The spatial attention map is then computed as follows:

$$M_s(F) = \sigma(f[F_{avg,s}; F_{max,s}]) \tag{11}$$

where f is the convolution operation $F_{avg,s} \in \mathbb{R}^{1 \times H \times W}, F_{max,s} \in \mathbb{R}^{1 \times H \times W}$.

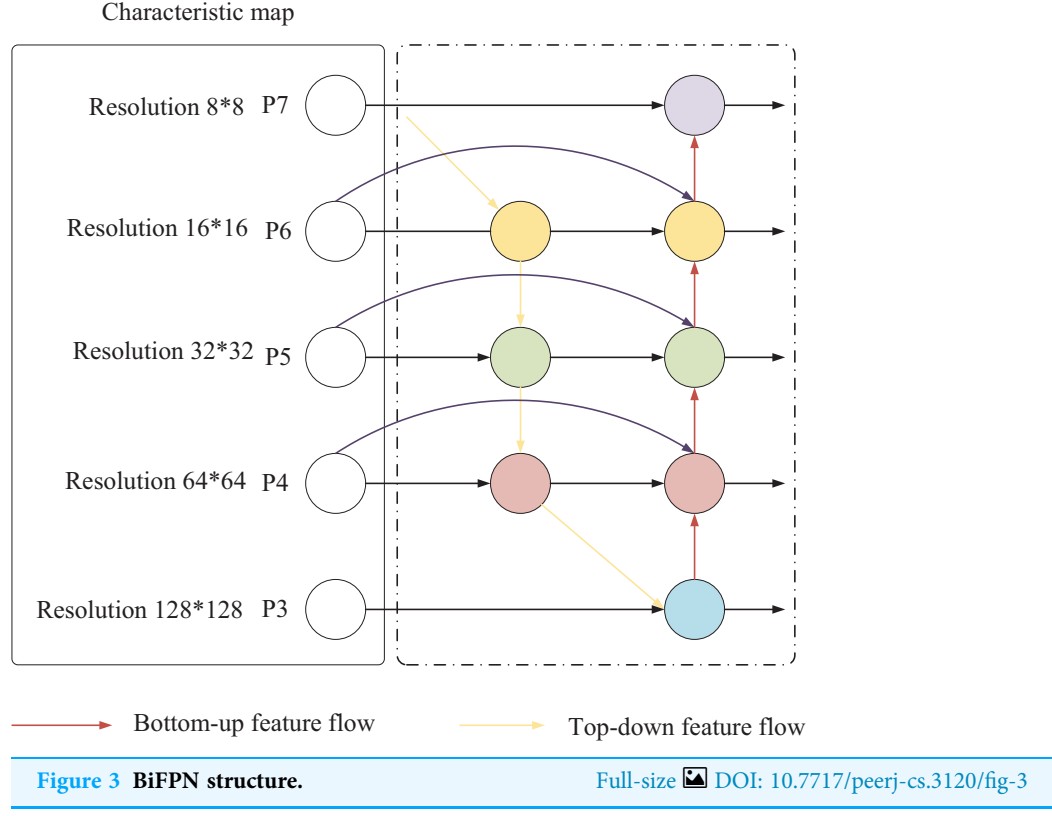

Characteristic map

Resolution 8*8  P7

Resolution 16*16  P6

Resolution 32*32  P5

Resolution 64*64  P4

Resolution 128*128  P3

→ Bottom-up feature flow          → Top-down feature flow

**Figure 3** **BiFPN structure.**     

## ML-BiFPN model

The traditional YOLO backbone network primarily employs the Focus structure and CSP structure. To address the challenge of limited small-sample data, SwinTransformer employs a sliding window mechanism and hierarchical design to enhance feature extraction capabilities for small samples. Additionally, in traditional YOLO, both the feature pyramid network (FPN) and PANet neck network structures contain nodes with only one input edge, which introduces excessive parameters and computational complexity. Furthermore, these nodes generally contribute less to feature fusion within the network. To mitigate this issue, this article adopts the BiFPN structure, as shown in Fig. 3. Notably, this study primarily focuses on the fusion of audio and image data to explore its applications in market forecasting and decision-making in international trade. Unlike conventional feature fusion approaches, BiFPN simplifies the network by removing nodes with only one input edge, streamlining the bidirectional network structure. The original input and output nodes are placed within the same layer, and an additional edge is added to facilitate the fusion of more features without significantly increasing computational load.

This bidirectional information flow facilitates communication between features at different scales, allowing for the effective integration of high-level semantic information with low-level, detailed information. Each fusion node in the BiFPN structure employs a weighted summation to combine various features, including those from the same scale in

**Table 1 Parameter setting.**

| Hyperparameter | Value |
| --- | --- |
| Learning rate | 0.001 |
| Optimizer | AdamW |
| Regularization | L2 regularization, coefficient 0.01 |
| Dropout | 0.2 |
| Batch size | 32 |
| Epochs | 100 |

the previous layer, higher-scale features obtained through upsampling, and lower-scale features acquired through downsampling. This weighted summation approach enables the model to adaptively adjust the importance of different source features based on the training data. The BiFPN employs a fast normalized fusion weighting method to assign appropriate weights to each input feature, which is calculated as follows:

$$O = \sum_i \frac{w_i}{\varepsilon + \sum_j w_j} \tag{12}$$

the activation function rectified linear unit (ReLU) is for each $w_i$ added to ensure that $w_i >$ 0. In the BFPN structure of Fig. 2, the expression of the fusion process of the two features formed by the $P_6$ node is as follows:

$$P_6 = Cov\left(\frac{w_1 \cdot P_6^{in} + w_2 \cdot \mathrm{Resize}(P_7^{in})}{w_1 + w_2 + \varepsilon}\right) \tag{13}$$

$$P_6^{out} = Conv\left(\frac{w_1' \cdot P_6^{in} + w_2' \cdot P_6 + w_3' \cdot \mathrm{Resize}(P_5^{out})}{w_1' + w_2' + w_3' + \varepsilon}\right) \tag{14}$$

where $P_6$ is the intermediate result of layer 6 in the top-down path, and $P_6^{out}$ is the output result of layer 6 in the bottom-up path; *Resize* is the resolution-assigned upsampling or downsampling operation, and *Conv* is the convolution operation.

## Computing infrastructure

The experiments were conducted on a high-performance computing setup running Ubuntu 20.04 LTS as the operating system. The hardware configuration consisted of an Intel Core i9-12900K CPU, 64 GB of RAM, and an NVIDIA RTX 3090 GPU with 24 GB of VRAM. The deep learning models were implemented using PyTorch 1.12 with CUDA 11.6 support for GPU acceleration. Meanwhile, the parameter settings are as shown in Table 1.

### 3rd party dataset

The public dataset used is Trade Map, a global trade data platform developed by the International Trade Centre, providing detailed import and export data for countries worldwide. This dataset includes images and audio related to the products (Dataset: https://zenodo.org/records/14035290).

*Selection method*

The techniques used in this study were selected based on a thorough literature review of state-of-the-art methods in multimodal learning and feature fusion. MFCC was chosen for audio due to its superior performance in capturing perceptual characteristics of speech. CBAM was selected for its lightweight yet effective attention modeling. ML-BiFPN was adopted and enhanced with meta-learning principles because of its strong ability to aggregate multi-scale features with bidirectional flows. The combination was specifically designed to address the complexity and heterogeneity of international trade data.

## EXPERIMENTS AND ANALYSIS

In this section, we analyze the proposed multimodal data fusion model within the framework of meta-learning. We evaluate the model's performance in feature extraction and feature fusion, with a focus on its ability to provide accurate decision-making. Additionally, we assess the model's effectiveness in predicting international trade market data, highlighting its capability to integrate and process multimodal inputs for improved prediction accuracy.

### Experimental environment and data set

We utilize meta-training to integrate multimodal data and assess the prediction performance on the corresponding publicly available dataset. During the meta-training phase, we set the batch size to 16 and trained for 100 epochs, with 200 iterations per epoch. The Mosaic data augmentation parameters follow those of the original YOLOv5, and the size of the convolution kernel $k$ is set to 5 in the NAM network. The public dataset used is the Trade Map, a global trade data platform developed by the International Trade Center (ITC). The Trade Map provides detailed import and export data for countries worldwide, including data related to goods such as images and audio.

### Evaluation criteria

To evaluate the performance of audio data and image data processing more fairly and accurately, the mean average precision (AP) of accuracy is used as the evaluation index for the algorithms in the testing stage. The higher the value of mean AP, the better the performance of the audio data processing model. The formula for calculating AP is as follows.

$$AP = \frac{1}{R}\sum_{i=1}^{R} P(i). \tag{15}$$

To evaluate the performance of audio data and image data processing more fairly and accurately, the AP is used as the evaluation metric in the testing stage. The higher the value of AP, the better the performance of the audio data processing model. The formula for calculating AP is as follows:

$$Pr\,ecision = TP/(TP + FP) \tag{16}$$

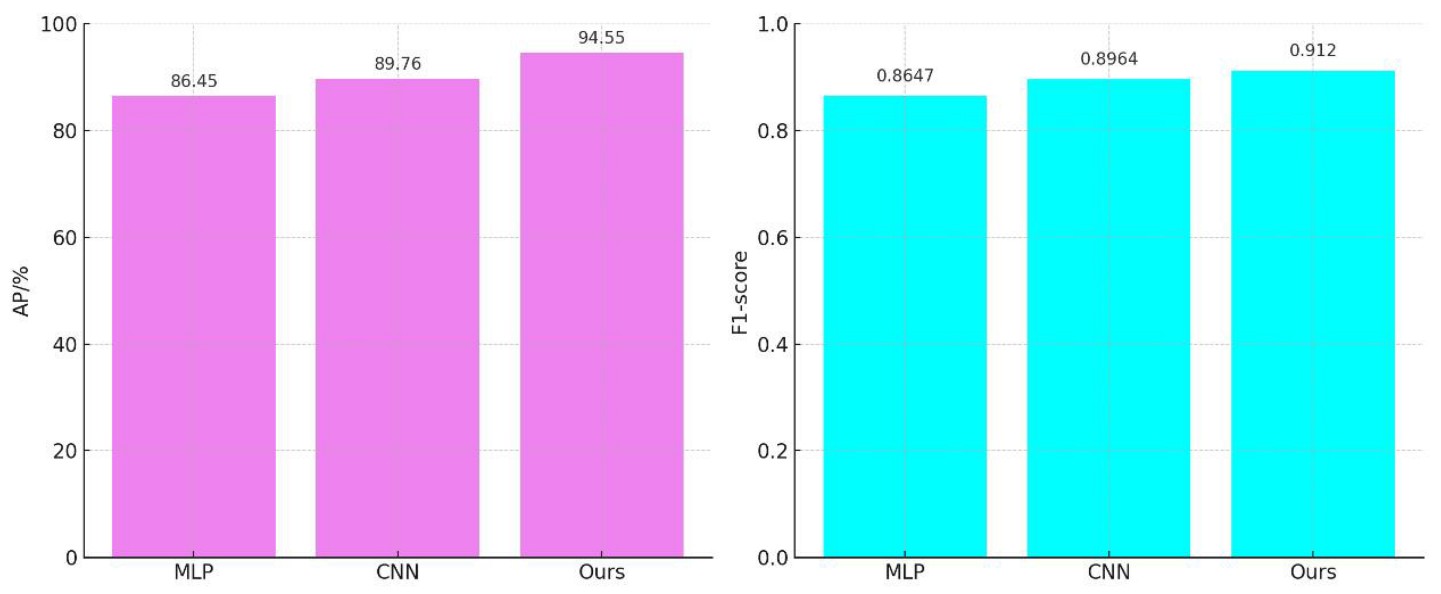

**Figure 4** Comparison of audio feature extraction performance.

$$Recall = TP/(TP + FN) \tag{17}$$
$$F1\text{-}score = 2 \times (Precision \times Recall)/(Precision + Recall). \tag{18}$$

## Performance analysis of feature processing

The results of the comparative audio feature processing are shown in Fig. 4. The average accuracy of the MLP-based feature extraction model (*Kang et al., 2022*) is 86.45%, with an F1-score of 0.8647. The convolutional neural network (CNN)-based feature extraction model (*Sun et al., 2022*) improves the average accuracy by 3.83% and the F1-score by 0.0317 compared to the MLP-based feature extraction model. Our audio feature extraction model further enhances the average accuracy by 9.37% and the F1-score by 0.0473 compared to the MLP-based feature extraction model.

Unlike the MLP model, which learns the correlation between the input and output through the hidden layers, the weight parameters in the CNN model are connected to only a small region of the input, corresponding to the location of the input rather than to the entire input. This local connection pattern enables the neurons to perceive only local information, which is then merged at higher levels to form a more comprehensive representation of the audio and global information. As a result, the CNN model exhibits a stronger correspondence to the local features of the input.

From the data in Fig. 5, it can be seen that during image feature extraction, the average accuracy of the CNN-based feature extraction model is 76.28%, with an F1-score of 0.7692. The average accuracy and F1-score of the VGG16-based feature extraction model (*Yang et al., 2023*) and the ResNet50-based feature extraction model (*Sharma et al., 2023*) have improved. However, these models are susceptible to interference, as some images in the trade market contain only partial representations of the subject.

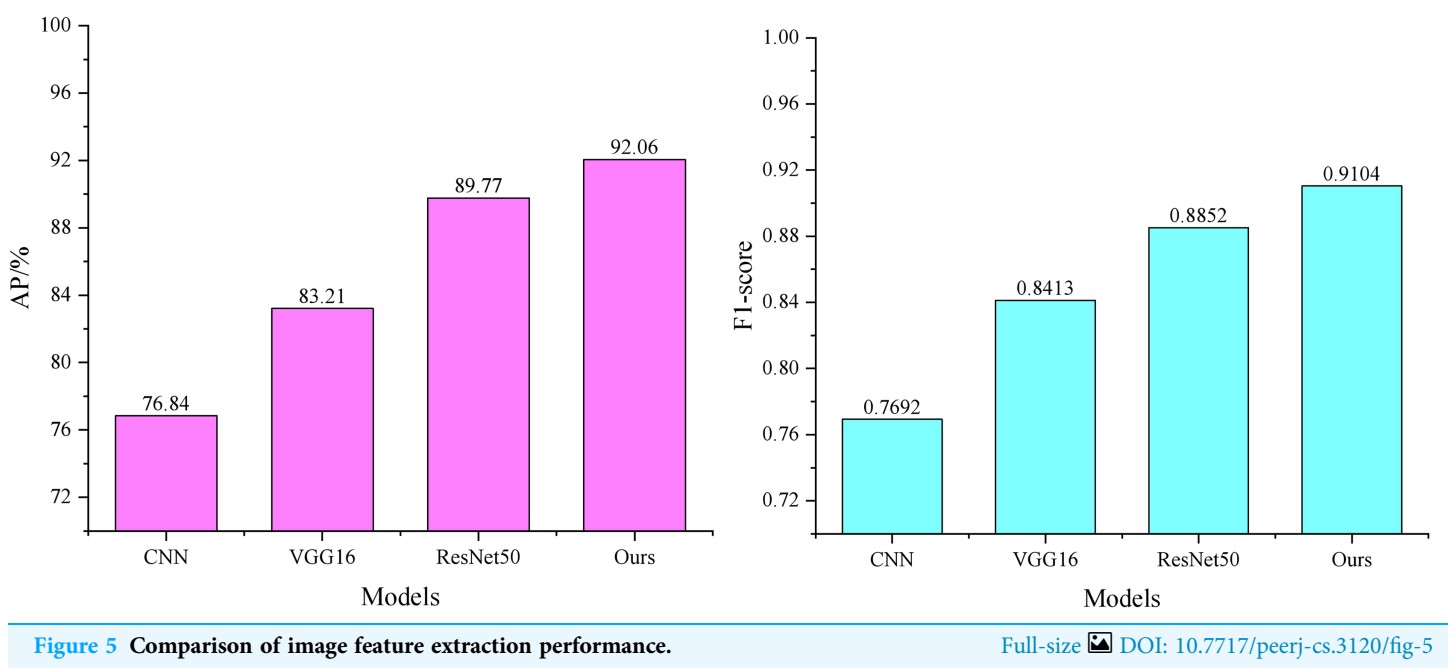

**Figure 5 Comparison of image feature extraction performance.**

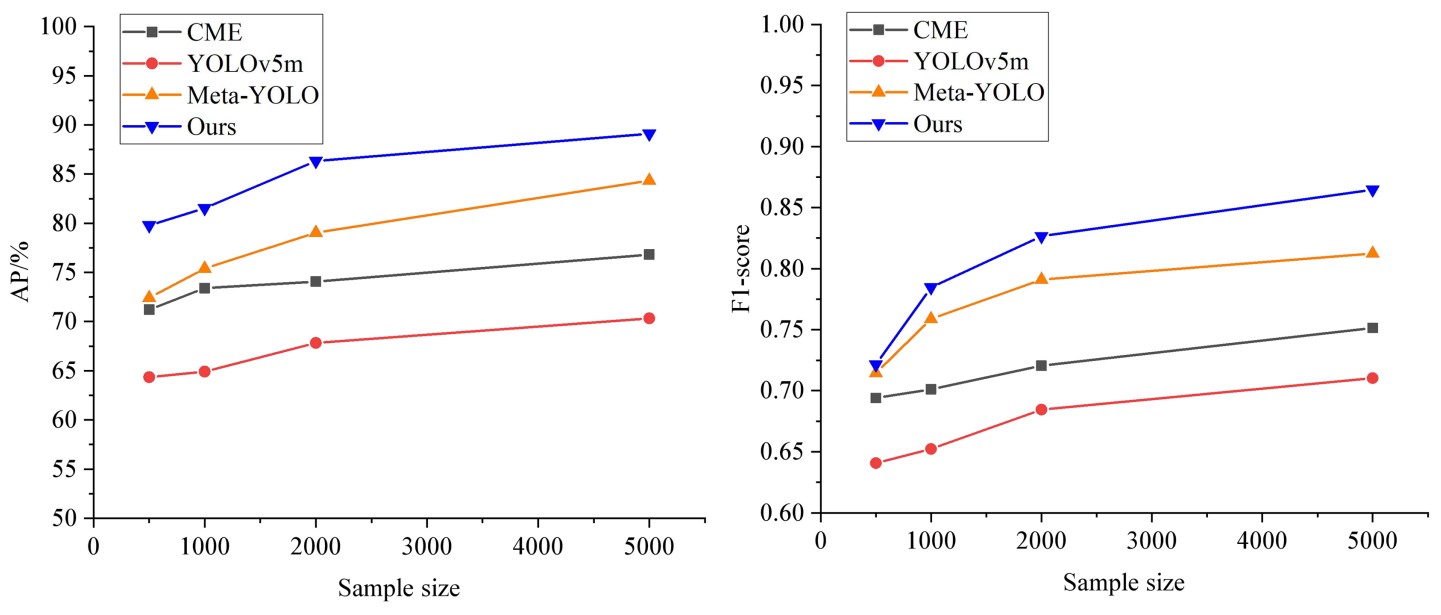

**Figure 6 Comparison of prediction performance among different models.**

In this article, the channel and spatial attention mechanism is introduced in the form of CBAM. In this context, the CAM weights the feature maps of each channel to enhance attention on the essential channels. In contrast, the SAM weights different regions of the feature maps to improve attention to crucial areas. The combination of these two modules allows the model to focus on both channel and spatial information simultaneously. This

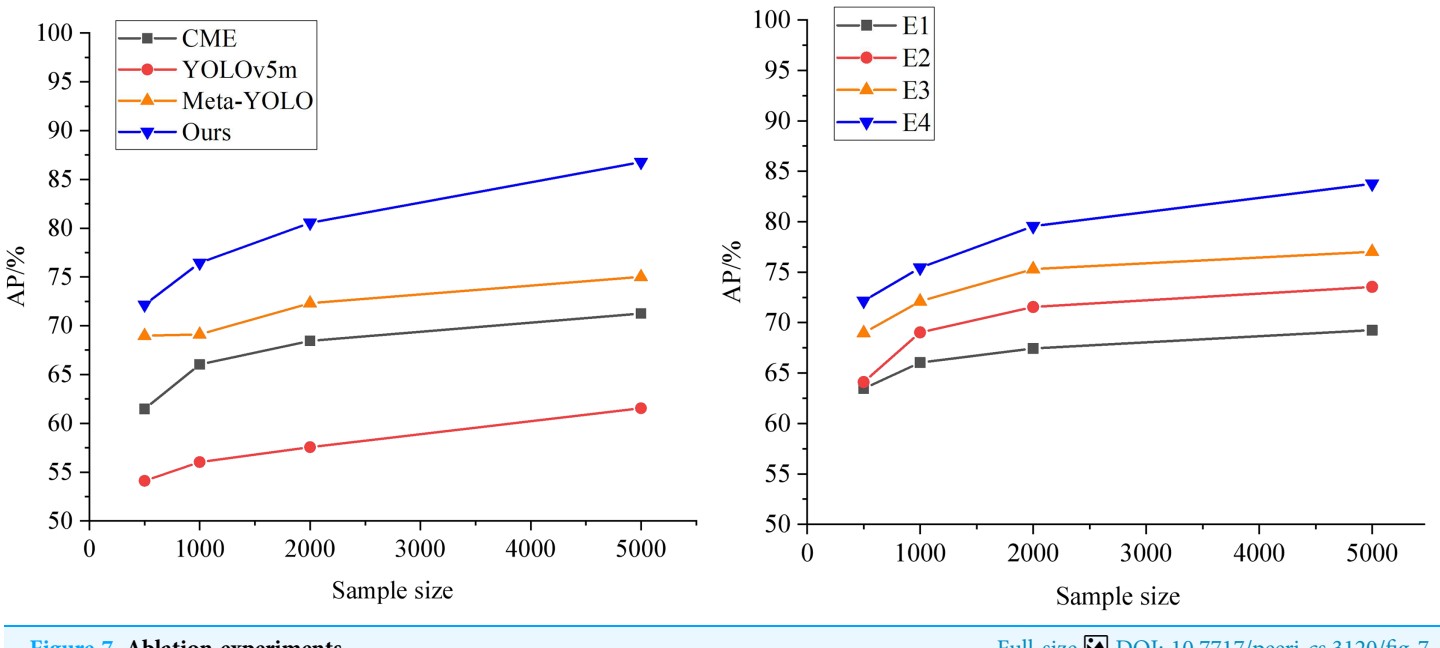

**Figure 7 Ablation experiments.**

enhances the feature representation capability of the ResNet50 model, yielding the best recognition performance with an average accuracy of 92.06%.

## Performance analysis under multimodal data fusion

After meta-training, the results are shown in Fig. 6. Among all the models, Meta-YOLO (*Lee et al., 2024*) demonstrates lower prediction accuracy due to its more straightforward feature extractor setup. YOLOv5m (*Dai et al., 2023*) also exhibits poor detection accuracy under low sample conditions. In contrast, CME (*Akinrinade, Du & Ajila, 2022*) shows relatively better results within the meta-learning framework. The AP value of the model proposed in this article is improved by an average of 5.13%, achieving the highest AP compared to other models, which indicates that the model maintains high prediction accuracy under conditions with small sample sizes. Additionally, the F1-score exhibits a smoother curve and better performance. The improved prediction accuracy of the ML-BiFPN model can significantly assist enterprises in more accurately grasping market dynamics, optimizing supply chain management, identifying market opportunities, and enhancing risk management capabilities. This, in turn, enables more scientific and informed decision-making in the international trade market.

To further analyze the role of the meta-learning module in the prediction model, ablation experiments are conducted by removing the meta-learning framework from each comparison model. The results are shown in the left graph of Fig. 7. It is evident that when YOLOv5m does not utilize the meta-learning framework, its base class performs relatively poorly. Meta-YOLO, with its weak feature extractor, shows poor detection accuracy. CME, on the other hand, maps features of different sample categories into distinct feature spaces, expanding the Euclidean distance between categories, which leads to better detection
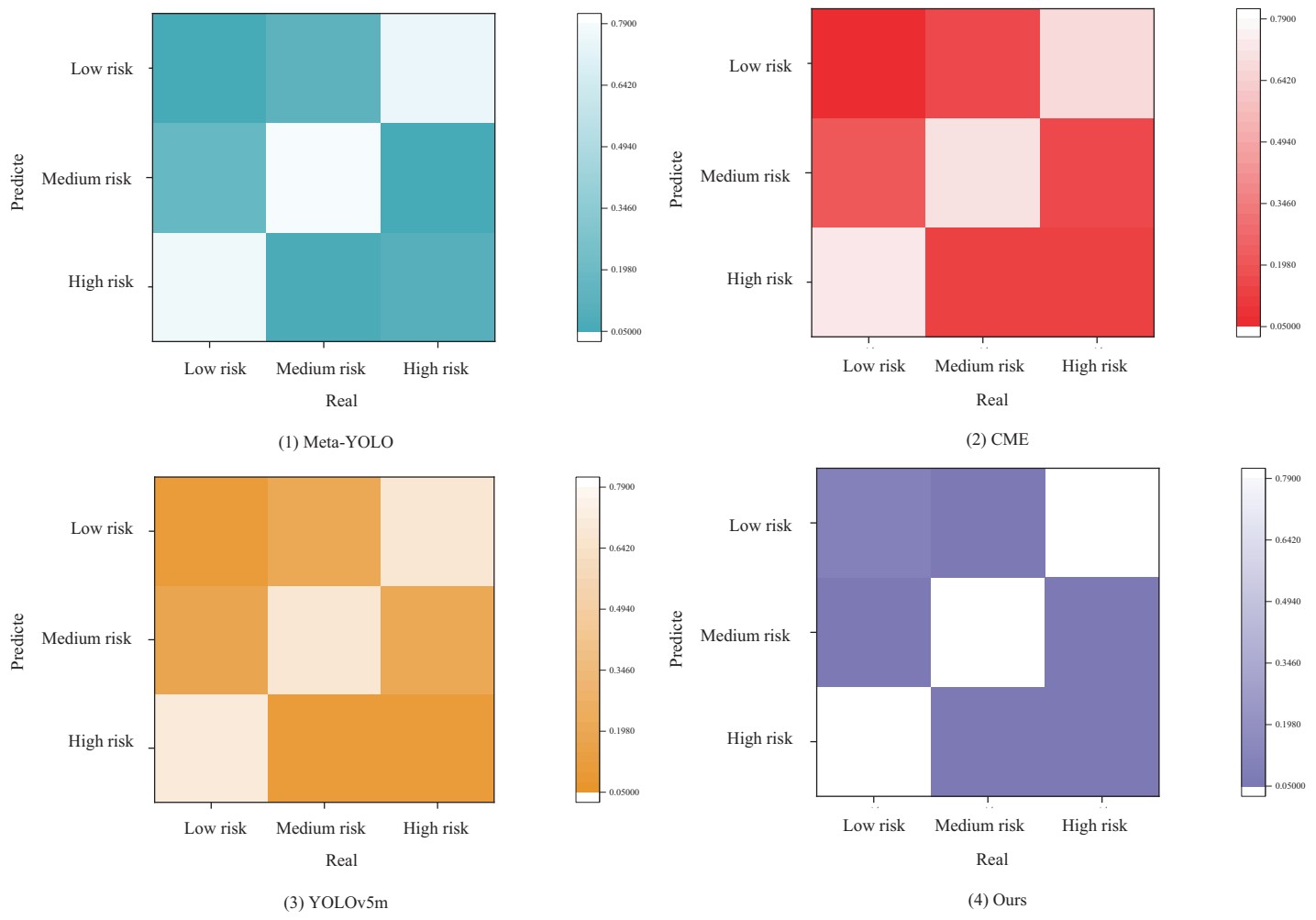

**Figure 8 Confusion matrix.**

accuracy. The model presented in this article achieves the highest AP across all sample sizes at this stage. Compared to the original Meta-YOLO, the average AP improvement rate of the ML-BiFPN model across all small sample sizes is 2.79%.

Furthermore, we sequentially remove the meta-learning framework, the BiFPN, CMAM, and MFCC modules from the ML-BiFPN model to obtain the corresponding experimental models (E4, E3, E2, and E1), as shown in the right panel of Fig. 7. It is clear that the ML-BiFPN model outperforms in multimodal feature fusion. The addition of the SwinTransformer Block and BiFPN backbone network significantly improves the model's feature extraction capabilities. However, when the BiFPN module is removed, the AP performance decreases relatively quickly, indicating its importance in enhancing the overall performance.

Finally, we analyze the results of each prediction model for risk assessment based on the confusion matrix. To assess the robustness of the proposed the meta-learning bidirectional feature pyramid network (ML-BiFPN), ablation experiments were conducted. Different

components, including the CBAM attention mechanism and the adaptive weighting strategy, were selectively removed or replaced to evaluate their contributions. These experiments helped isolate the impact of each architectural decision on model performance.

As shown in the confusion matrix in Fig. 8, the models predict three categories: low risk, medium risk, and high risk. The definition and labeling of risk categories are grounded in the analysis of historical transaction data and the judgment of domain experts.

To begin with, we segment different risk levels based on key indicators within historical transaction data, such as transaction amount, transaction frequency, and the creditworthiness of trading counterparts. Subsequently, leveraging the knowledge and experience of domain experts, we refine and adjust these risk levels further.

When setting thresholds for low-, medium-, and high-risk transactions, we adopt a hybrid approach that combines statistical methods with domain expertise. For instance, transactions with amounts below a certain threshold can be classified as low-risk; those within a specific range can be categorized as medium-risk; and transactions exceeding a particular threshold can be labeled as high-risk.

From Fig. 8 ((1) Meta-YOLO), it is evident that the Meta-YOLO model performs moderately well in predicting low and medium risks; however, its prediction accuracy for high risks is relatively poor and requires improvement. In Fig. 8 ((2) CME), the match between high-risk predictions and actual high-risk instances in the CME model is not sufficiently high, indicating some errors in identifying high-risk categories. In sub-figure (3), the YOLOv5m model performs clearly in predicting low and high risks, but its prediction of medium risk is somewhat ambiguous. Additionally, its overall risk prediction and assessment performance are lower compared to other models.

In comparison, the ML-BiFPN model proposed in this article outperforms all three risk categories, demonstrating superior accuracy in risk prediction. This makes the ML-BiFPN model a valuable tool for guiding investment decisions in the trade market.

## CONCLUSION AND LIMITATIONS

In this article, key technologies in the international trade market prediction and decision-making system are discussed in depth, with a particular focus on the method of realizing multimodal data fusion under the meta-learning framework. A YOLO framework based on meta-learning is proposed, enabling the fusion of multimodal data, such as audio and image, under this framework. Through experimental analysis, the ML-BiFPN model proposed in this article demonstrates excellent performance in feature extraction and feature fusion, significantly improving market prediction accuracy and the scientific validity of decision-making. Experimental results on the publicly available Trade Map dataset demonstrate that the model's AP value is improved by an average of 5.13%, resulting in higher prediction accuracy compared to other models. This article represents significant progress in the fields of multimodal data fusion and meta-learning. The model proposed in this study integrates multiple resource-intensive components, including SwinTransformer, BiFPN, and CBAM. As a result, it demands high computational

resources during both training and inference processes. Under the same hardware environment, the inference time for an audio clip varies depending on its length.

To meet the requirements of real-time trading systems, we are currently exploring model compression and acceleration techniques, such as quantization and pruning.

Despite its promising results, the study has several limitations:

**Generalizability across domains:** The model is evaluated solely on the Trade dataset. Its performance on other types of multimodal economic or financial datasets remains unexplored.

**Computational complexity:** Although the proposed architecture is effective, it is computationally intensive and may not be suitable for real-time deployment in resource-constrained environments.

**Absence of real-time testing:** The system has not been evaluated in a live trading environment, which may present unforeseen challenges such as latency, data noise, and user interaction.

Future work will address these limitations by extending the model to additional datasets, optimizing it for real-time applications, and incorporating explainable AI components for better interpretability.

### Funding

The authors received no funding for this work.

### Competing Interests

Muhammad Asif is an Academic Editor for PeerJ

### Author Contributions

- Yiming Bai conceived and designed the experiments, performed the experiments, analyzed the data, performed the computation work, prepared figures and/or tables, authored or reviewed drafts of the article, and approved the final draft.
- Muhammad Asif conceived and designed the experiments, performed the experiments, analyzed the data, prepared figures and/or tables, authored or reviewed drafts of the article, and approved the final draft.

### Data Availability

Data is available at Zenodo:

Bavaresco, A., Testoni, A., & Fernández, R. (2024). TRADE [Data set]. Association for Computational Linguistics (ACL), Bangkok, Thailand. Zenodo. https://doi.org/10.5281/zenodo.14035290.

Code is available in the Supplemental Files.

## Supplemental Information

Supplemental information for this article can be found online at http://dx.doi.org/10.7717/peerj-cs.3120#supplemental-information.

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
