# Peer review of "International trade market forecasting and decision-making system: multimodal data fusion under meta-learning"

_PeerJ Computer Science, doi:10.7717/peerj-cs.3120_

## Round 0.1 · original submission · Major Revisions

Please revise your paper based on the three sheets of comments.

·

Basic reporting

1. Clarify how the use of audio and image data maps to real trade decision-making processes. Are these voice calls, meeting recordings, and product photos common in actual trade workflows?
2. The inclusion of audio and image modalities needs stronger justification in terms of real-world application. For example, explain the relevance and availability of voice call data and product images in trade decision-making.
3. While text preprocessing is briefly mentioned, the model architecture doesn’t appear to integrate textual inputs. Clarify or remove to avoid confusion.
4. There are several grammatical issues (e.g., missing articles, awkward phrasing like “enhance the model9s focus”) that detract from readability. Please revise with a professional language edit.
5. The current narrative jumps between concepts (e.g., unidimensional data, multimodality, meta-learning) without sufficient transition. Try organizing the introduction into three clear parts: (1) background/problem, (2) limitations of current approaches, and (3) the proposed solution and contributions.
6. The literature review heavily cites works from vision, audio, and sentiment analysis. It should include more references relevant to economic forecasting, international trade analytics, or AI in business decision-making to better ground the study in its application domain.

Experimental design

1. The paper mentions the Trade Map dataset but does not specify how multimodal elements (e.g., audio, image) were generated or labeled. This is essential for reproducibility.
2. Include a table summarizing key hyperparameters for reproducibility (e.g., learning rate, optimizer, regularization, dropout rate).
3. The model includes multiple components (SwinTransformer, BiFPN, CBAM), which are resource-intensive. Discuss the training and inference time and potential deployment challenges, especially in real-time trade systems.
4. YOLO is primarily a detection model. Since the task here is prediction and decision-making, clarify how YOLO is adapted and applied in this context.

Validity of the findings

1. The paper uses a meta-learning framework but does not deeply contrast it with other meta-learning approaches. Include discussion on why your setup is particularly suitable for trade forecasting.
2. Acknowledge trade-offs between accuracy and inference cost. Could a lighter version of the model be proposed?
3. Figure 8 introduces risk classes. Explain how these were defined and labeled (e.g., thresholds for low, medium, high risk).
4. The current limitations section is useful but could be enhanced by including risks like data bias, privacy concerns, and robustness to noisy input.

Additional comments

There are numerous grammatical issues (e.g., “the model enhances its focus on crucial channels” → "the model enhances focus on salient channels"). A professional edit is recommended.

Reviewer 2 ·

Basic reporting

The authors present a multimodal data fusion approach using Meta-Learning for international trade market forecasting. The proposed ML-BiFPN model leverages audio (MFCC) and image (CBAM) features, showing improved performance over other methods, particularly with small sample sizes. The manuscript is generally well-structured and addresses an interesting and relevant problem. However, there are several critical and minor points that need to be addressed before it can be considered for publication.

Experimental design

- The paper claims to leverage audio, image, and structured data modalities. However, the exact nature of these inputs is unclear. For instance, what specific audio content was used (e.g., trade meeting recordings, media broadcasts)? What type of images were analyzed (e.g., product images, site inspection photos)? A table summarizing the modalities, sample counts, input formats, and corresponding labels would significantly improve transparency.

- To promote reproducibility, it is recommended that the authors provide either preprocessing scripts or detailed metadata about the dataset structure, preprocessing steps, and how the raw data was transformed into training inputs. Although a code link is provided, the manuscript lacks sufficient detail regarding how the preprocessing was performed and how different modalities (e.g., audio, image, and structured data) were aligned and integrated for model training.

- The paper refers to risk classification (low, medium, high) but does not explain how these labels were generated. Were they manually annotated, derived from structured trade indicators, or algorithmically assigned? Furthermore, there is no information about the distribution of samples across these classes. Please provide the class balance statistics and justification for label definitions.

Validity of the findings

- The manuscript references both the "TRADE" dataset (TRuly ADversarial ad understanding Evaluation) and the "Trade Map" dataset by the International Trade Center (ITC). These are fundamentally different datasets, and it is unclear which one was actually used. Please clarify which dataset forms the basis of the experimental analysis and ensure consistency throughout the manuscript.

- The manuscript lacks a clear description of the train/validation/test split. It is not mentioned whether cross-validation was performed or whether fixed splits were used. Additionally, the absence of repeated experiments or reporting of standard deviations casts doubt on the statistical robustness of the reported performance.

Reviewer 3 ·

Basic reporting

The reporting of the paper is quite hard to follow. The english is, in fact good but the introduction needs to have an ending where it introduces the following sections.

The related work needs to be more focused on the multimodal trade market forecasting problem. What other research addressed this topic from the machine learning point of view? What datasets are available? What did others achieve on such datasets?

Experimental design

The materials and methods section is very hard to follow. I would recommend having one figure that combines all modalities to help the readers visualise what you are proposing.

Various equations were mentioned throughout the sections yet without any references.

The subsection named "data pre-processing" is part of the proposed model and should have been included in the overall structure of the overall model.

The evaluation methods section is not needed but rather this text is to be moved to the experimental (ablation study part)

The assessment metrics should have the explanation of the metrics which you placed in the subsection 4.2 (why repeat subsections)

The dataset was hardly presented; only the name and link were given. How many modalities? How many records per modality? You need to elaborate on the dataset used.

Validity of the findings

The experimentation targeted each component independently. One for image, another for audio, then an ablation. I understand that you are reasoning the choice of each block in the proposed system but I can hardly validate the overall performance. Also, where is the comparison to other methods? How is figure 7 an ablation study? The ablation study involves proving that each modality is important in the overall performance of the proposed model.
The abstract mentioned an improvement of 9% compared to MLP. Isn't that for the audio feature only not the proposed model?? That is a very strong statement that was not supported in your results.

Additional comments

Overall, the idea is good but lacks a proper explanation of the overall proposed model and many experiments to better validate the results and compare them to others. The abstract is very misleading.

---

## Round 0.2 · accepted · Accept

The reviewers are satisfied that you have addressed their concerns and the manuscript is accepted.

Reviewer 2 ·

Basic reporting

The Introduction clearly motivates the need for multimodal data fusion and meta-learning in the context of international trade forecasting. Relevant prior literature is well-cited, and related works are discussed comprehensively.

Experimental design

- The study presents a rigorous methodology using MFCC for audio, CBAM for image data, and a meta-learning-enhanced BiFPN architecture.
- Data preprocessing is described in adequate detail, including anonymization, labeling, and dataset splits.

Validity of the findings

- The results are thoroughly analyzed with both performance metrics and visualizations (Figures 4–8).
- The conclusion logically follows from the experiments, including detailed ablation studies.

Additional comments

The authors have appropriately addressed my previous concerns.

Reviewer 3 ·

Basic reporting

The manuscript has been improved. My comments have been met.

Experimental design

The manuscript has been improved. My comments have been met.

Validity of the findings

The manuscript has been improved. My comments have been met.